# Association of Endotoxemia with Low-Grade Inflammation, Metabolic Syndrome and Distinct Response to Lipopolysaccharide in Type 1 Diabetes

**DOI:** 10.3390/biomedicines11123269

**Published:** 2023-12-10

**Authors:** Aleksejs Fedulovs, Leonora Pahirko, Kaspars Jekabsons, Liga Kunrade, Jānis Valeinis, Una Riekstina, Valdis Pīrāgs, Jelizaveta Sokolovska

**Affiliations:** 1Faculty of Medicine, University of Latvia, LV-1004 Riga, Latvia; aleksejs.fedulovs@lu.lv (A.F.); kaspars.jekabsons@lu.lv (K.J.); liga.kunrade@lu.lv (L.K.); una.riekstina@lu.lv (U.R.); valdis.pirags@lu.lv (V.P.); 2Faculty of Physics, Mathematics and Optometry, University of Latvia, LV-1004 Riga, Latvia; leonora.pahirko@lu.lv (L.P.); janis.valeinis@lu.lv (J.V.); 3Pauls Stradins Clinical University Hospital, LV-1002 Riga, Latvia

**Keywords:** type 1 diabetes, metabolic syndrome, endotoxin, low-grade inflammation, lipopolysaccharide-binding protein, endogenous anti-endotoxin core antibodies

## Abstract

The association of endotoxemia with metabolic syndrome (MS) and low-grade inflammation in type 1 diabetes (T1D) is little-studied. We investigated the levels of lipopolysaccharide (LPS), lipopolysaccharide-binding protein (LBP), endogenous anti-endotoxin core antibodies (EndoCAb IgG and IgM) and high-sensitivity C-reactive protein (hsCRP) in 74 T1D patients with different MS statuses and 33 control subjects. Within the T1D group, 31 patients had MS. These subjects had higher levels of LPS compared to patients without MS (MS 0.42 (0.35–0.56) or no MS 0.34 (0.3–0.4), *p* = 0.009). MS was associated with LPS/HDL (OR = 6.5 (2.1; 20.0), *p* = 0.036) and EndoCAb IgM (OR = 0.32 (0.11; 0.93), *p* = 0.036) in patients with T1D. LBP (β = 0.30 (0.09; 0.51), *p* = 0.005), EndoCAb IgG (β = 0.29 (0.07; 0.51), *p* = 0.008) and the LPS/HDL ratio (β = 0.19 (0.03; 0.41, *p* = 0.084) were significantly associated with log-transformed hsCRP in T1D. Higher levels of hsCRP and EndoCAb IgG were observed in T1D compared to the control (*p* = 0.002 and *p* = 0.091, respectively). In contrast to the situation in the control group, LPS did not correlate with LBP, EndoCAb, leukocytes or HDL in T1D. To conclude, endotoxemia is associated with low-grade inflammation, MS and a distinct response to LPS in T1D.

## 1. Introduction

The prevalence of autoimmune conditions, including type 1 diabetes (T1D), has markedly increased in the recent decades [1]. T1D is characterized by absolute insulin deficiency caused by the loss of insulin-producing pancreatic beta cells and increased morbidity and mortality mainly due to long-term diabetes complications [2]. It has been shown that metabolic syndrome (MS)—a combination of cardiovascular risk factors [3] and a state of low-grade inflammation [4]—is associated with the progression of both micro- and macrovascular complications of diabetes [5,6,7] in T1D. Increased bacterial lipopolysaccharide (LPS, endotoxin) leakage into the circulation, or endotoxemia, is considered to be a possible source of the chronic inflammatory reaction in T1D [8] and is associated with the progression of diabetic kidney disease (DKD) [9,10], cardiovascular disease (CVD), atherosclerosis [11] and MS components [12]. In addition to possible factors predisposing T1D patients to increased endotoxemia and low-grade inflammation (such as leaky gut syndrome, intestinal microbiota dysbiosis, hyperglycemia [8,13,14,15,16], etc.), impaired response to infection is of particular interest [17].

In addition to LPS activity itself, the serum concentration of its binding proteins such as LPS-binding protein (LBP) and the endogenous anti-endotoxin core antibodies (EndoCAb) IgM and IgG can serve as markers of infection severity and the host’s anti-inflammatory response [18,19]. Unfortunately, only a few studies report concentrations of some of these markers in T1D compared to healthy subjects [20,21,22] and in relation to MS components in T1D [23]. Also, no study to date reports correlations of the levels of fecal calprotectin as a marker of intestinal inflammation [15] with endotoxemia markers in T1D.

Finally, the potential link of non-alcoholic fatty liver disease (NAFLD) or its surrogate markers with endotoxemia in T1D has not yet been studied. This gap in knowledge should also be filled, as there is evidence of links between MS, NAFLD [24] and endotoxemia in the general population [25]. The latter data would be of high importance due to the increasing prevalence of NAFLD in T1D and its association with the complications of T1D [26,27]. 

To summarize, a better understanding of associations between low-grade inflammation, endotoxins and their binding proteins and MS components is important for the development of future treatment and prevention strategies for vascular T1D complications. 

In this work, we utilized several novel approaches of statistical analysis in addition to conventionally used methods; for example, for data normalization between the plates, a transformation based on the two-sample location-scale model was used [28,29]. For two-distribution comparisons, we used the smoothed empirical likelihood method for the probability–probability (PP) plot [30,31,32]. In addition, the empirical likelihood method for the equality test of all deciles of two distributions, which allows for obtaining information about the differences or similarities of two samples, was applied for data analysis [33].

To conclude, the aim of this study was to investigate the levels of endotoxins and their binding proteins in T1D patients with different MS statuses and to study associations between endotoxemia markers, low-grade inflammation and MS using both conventional and novel approaches of statistical analysis. 

## 2. Materials and Methods

### 2.1. Study Groups and Subjects

All participants in this investigation took part in the longitudinal LatDiane study, which is a part of the international InterDiane consortium. Adult patients with T1D diagnosed before the age of 40 years, with insulin treatment initiated within one year of diagnosis and C-peptide levels below 0.3 nmol/L, were recruited for LatDiane. Patients with a history of chronic kidney disease apart from DKD were excluded [34]. Follow-up visits and re-assessment of the status of complications of diabetes were performed every three years or more frequently.

We performed recruitment, biobanking and sample storage in agreement with the procedures of the Genome Database of the Latvian Population [35,36].

Recruitment took place between 15th January 2021 and 31st August 2021 in Latvia, Riga, Pilsoņu 13 str. building 10 (the rooms of the Laboratory for Personalized Medicine of the University of Latvia). The group of subjects with normal glucose metabolism (control group) included non-smoking spouses and friends of patients participating in this study, university personnel and students willing to participate in this study. Information about the study was disseminated via a webpage and social media of the University of Latvia.

The inclusion criterion for the group of patients with T1D was a diabetes duration of at least 8 years.

The inclusion criteria for the control group were normal glucose metabolism, defined as fasting glucose ≤ 5.6 mmol and HbA1c ≤ 5.7% [2], and no known or documented autoimmune, cardiovascular or other chronic diseases.

Exclusion criteria for both study groups were pregnancy, history of inflammatory bowel disease (Crohn’s disease or ulcerative colitis) and coeliac disease, acute intestinal infection within 2 months of the planned fecal collection, asymptomatic coeliac disease (detected via screening of serum transglutaminase IgA antibodies), clinical signs of acute inflammation, and fever.

On the study day, patients were investigated for the collection of anthropometric measures, blood samples and information on disease history. They also received instructions and vials for the collection of the fecal sample.

### 2.2. Clinical Definitions

Patients were investigated to document weight, height, waist circumference and blood pressure. The body mass index (BMI, weight (kg)/height (m)^2^) and waist/height ratio were calculated. Patients with systolic blood pressure ≥ 140 mmHg (18.7 kPa) or diastolic blood pressure ≥ 90 mmHg (12.0 kPa), or a history of antihypertensive drug usage, were defined to have arterial hypertension.

Smoking was self-reported, and the “smokers” group referred to patients currently smoking at least one cigarette per day.

Medical files were investigated for assessment of CVD and complications of diabetes such as retinopathy, neuropathy and DKD. We defined CVD as a history of acute myocardial infarction, coronary bypass/percutaneous transluminal coronary angioplasty stroke, amputation or peripheral vascular disease.

The albumin-to-creatinine ratio in two out of the three morning spot urine samples was used for the definition of albuminuria status. The estimated glomerular filtration rate (eGFR) was calculated according to the Chronic Kidney Disease Epidemiology Collaboration (CKD-EPI).

MS was assessed according to Alberti et al. [3]. The waist criterion was fulfilled for men if ≥102 cm and for women if ≥88 cm. Patients with serum triglycerides ≥ 1.7 mmol/L, serum high-density lipoproteins < 1.0 mmol/L (for men) and <1.3 mmol/L (for women) or medication to manage these dyslipidemia disorders fulfilled these respective criteria. Furthermore, subjects with blood pressure exceeding 130/85 mmHg or antihypertensive treatment fulfilled the blood pressure criterion. The elevated fasting blood glucose criterion was positive for all subjects with T1D. In the case of ≥3 positive criteria, MS was confirmed.

### 2.3. Sampling of Blood for Serum Preparation and Fecal Collection

Blood samples and morning spot urine samples were sent to a certified clinical lab for assessment of clinical markers (blood count, CRP, clinical chemistry, albuminuria).

For serum preparation for further analysis of endotoxemia markers and hsCRP, peripheral venous blood was collected. The blood samples were incubated undisturbed for 30 min at room temperature and then centrifuged. The serum was removed from the pellet and transferred into fresh 2 mL tubes, frozen and stored at −20 °C until analysis.

Participants collected their fecal samples at home within two weeks after blood collection, using sterile collection tubes without buffer (collection date and time were marked). Within 24 h, samples were delivered to the laboratory for calprotectin measurement in unfrozen samples.

### 2.4. Determination of Inflammatory and Endotoxemia Markers

In serum, the LPS activity was measured using a Hycult LAL chromogenic endpoint assay (HIT302, Hycult Biotech, Uden, The Netherlands). EndoCAb IgG and EndoCAb IgM were measured using Hycult EndoCAb IgG and IgM Elisa kits (HK504-IGG; HK504-IGM; Hycult Biotech, Uden, The Netherlands). LPB was measured using a Hycult LPB Human Elisa kit (HK315-02, Hycult Biotech, Uden, The Netherlands). hsCRP was measured using a Hycult Human hsCRP kit (HK369 Hycult Biotech, Uden, The Netherlands). Measurements were performed according to the manufacturer’s instructions. Fecal calprotectin was measured using an Alegria^®^ Calprotectin Elisa kit (REF ORG280, Organotech Diagnostika GmbH, Budapest, Hungary) in a certified clinical lab.

### 2.5. Indices of Insulin Resistance and NAFLD

Estimated glucose disposal rate (eGDR) allows for the estimation of insulin resistance in patients with T1D. It was calculated according to the following formula:eGDR=24.4−12.97×Waist/Hips−3.39×Hypertension−0.6×HbA1c%,
where hypertension is 1 if blood pressure is ≥140/90 mmHg and/or the patient takes antihypertensive drugs regularly; otherwise, it is 0 [37,38]. The lower the eGDR, the higher the insulin resistance.

The NAFLD hepatic steatosis index (HSI) and fatty liver index (FLI) were calculated according to the following formulas:HSI=8×ALT/AST+BMI+2 (if DM)+2(if female), 
with values < 30 ruling out and values ≥ 36 ruling in steatosis [39], and
FLI=logistic(0.953×ln(TG)+0.139×BMI+0.718+ln(GGT)+0.053×Waist−15.745)×100,
where $logistic\left(x\right)=1/\left(1+e^{-x}\right)$ denotes the logistic function and ln(x) the natural logarithm. Values < 30 rule out and values ≥ 60 rule in steatosis [40].

### 2.6. Statistical Analysis

All statistical analysis was performed using statistical open-source software R version 4.3.1 (http://www.r-project.org accessed on 20 September 2023) [41]. The Shapiro–Wilk test was used to test the normality assumption. We report all *p*-values below 0.1 as statistically significant throughout the paper.

#### 2.6.1. Data Normalization across the Plates

Data on five serum inflammatory markers were accessed on 1 February 2023. They were examined for homogeneity across three plates. The distributions were highly skewed to the left (the skewness coefficient ranged from 0.66 to 1.93); the normality assumption was violated for all five markers (*p* < 0.001). The equality of locations across three plates was tested using the Kruskal–Wallis test and was not rejected for hsCRP (*p* = 0.46), LPS (*p* = 0.24), EndoCAb IgG (*p* = 0.82) and EndoCAb IgM (*p* = 0.22). However, variability was detected for LBP measurements across the plates (*p* < 0.001) (Appendix A). Post hoc comparisons demonstrated that measurements of plate 3 differed significantly from both plate 1 and plate 2. It was decided to normalize observations of LBP via simultaneous location and scale transformation for plate 3 based on a two-sample location-scale model. We verified the simultaneous location-scale change between two distributions using the test developed by Hall et al. [29]. The hypothesis of location-scale change between plates 1 and 2 and plate 3 was not rejected (*p* = 0.76). To obtain normalized data, we transformed observations on plate 3 by multiplying with the scale estimate 6.475 and by adding the location estimate −24.83. The equality of locations across all the plates of normalized LBP was not rejected (Kruskal–Wallis *p* = 0.93).

#### 2.6.2. Descriptive Statistics and Comparisons between Two Samples

Categorical variables were summarized as counts and percentages. The equality of proportions between the two study groups was tested using the chi-square test for proportions. Most of the continuous variables analyzed violated the normality assumption; therefore, data are presented as medians with interquartile range (IQR). For descriptive statistics, we used several plots to compare the two samples: probability–probability plots, smoothed kernel density estimates, one-dimensional scatter plots and quantile difference plots. The pointwise confidence intervals were added for PP plots and quantile difference plots.

Comparisons between the two samples were performed by using the Kolmogorov–Smirnov, the Wilcoxon and the Lepage tests. The Wilcoxon test compares two medians, the Lepage test determines the statistically significant change in location and scale and the Kolmogorov–Smirnov test compares whole distributions. The Wilcoxon test was followed by ANCOVA on ranks to perform the covariate adjustment. Finally, the empirical likelihood method was used to test for equality of all deciles of two distributions. All tests were implemented in the statistical software R. Package EL was used to produce the empirical likelihood-based statistical inference.

#### 2.6.3. Correlations and Regression Analysis

Correlation analysis was carried out separately in the control and T1D groups. Spearman’s rank correlation coefficient ρ with a 95% confidence interval and the *p*-value for the significance of the association were calculated. The results are illustrated in a separate forest plot for each variable of interest.

A linear multivariate regression model was fitted on log-transformed hsCRP in the T1D group with LBP, EndoCAb IgG, EndoCAb IgM, LPS/HDL, sex, BMI and diabetes duration as the predictors (Model 1). Logistic multivariate regression was performed to predict the MS in T1D using LBP, EndoCAb IgG, EndoCAb IgM, LPS, hsCRP, sex, BMI and diabetes duration as the predictors (Model 2). All predictors were standardized before the model fitting in both cases. Multicollinearity was examined using variance inflation factor calculations. The R-squared and Nagelkerke R-squared were calculated to assess the fit for Model 1 and Model 2, respectively.

## 3. Results

### 3.1. Characteristics of Subjects

In this study, 74 patients with T1DM were included. In the T1D group, the median diabetes duration was 21 (13–32) years and 31 subjects had MS. Compared to patients without MS, subjects with MS had a higher BMI, waist/height ratio and prevalence of diabetic complications and used antihypertensive and lipid-lowering medications more frequently. In addition, patients with MS had lower insulin sensitivity as assessed using the eGDR formula, higher HbA1c and had higher scores of NAFLD surrogate markers FLI and HSI (Table 1).

We also included 33 generally healthy adults (control) in this study. A detailed comparison of the control vs. T1D group can be found in Appendix A. In brief, patients with diabetes and controls did not differ in anthropometric measures and gender distribution. Compared to subjects in the control group, patients with diabetes were older (control: 35 (30–44) years; T1D: 43 (34–51), *p* = 0.013); had higher HbA1c levels (control: 5.2 (5.0–5.5), T1D: 7.7 (6.9–9.3), *p* < 0.001); had a higher prevalence of MS (control: 4 (12.1%), T1D: 31 (41.9%), *p* = 0.005), hypertension (control: 7 (21.2%), T1D: 35 (27.0%), *p* = 0.019) CVD (control: 0 (0%), T1D: 9 (12.2%), *p* = 0.086) and autoimmune thyroid disease (control: 1 (3%), T1D: 19 (25.7%), *p* = 0.012); and had a higher level of albuminuria (control: 0.32 (0.11–0.76), T1D: 0.32 (0.11–0.76), *p* = 0.013) and lower level of insulin sensitivity as assessed using eGDR (control: 7.1 (5.5–7.7), T1D: 3.3 (1.6–5.7), *p* < 0.001).

### 3.2. Levels of Serum Inflammatory Markers in Patients with T1D Stratified According to MS Presence and Controls

Patients with T1D and MS had higher levels of LPS compared to patients without MS (MS: 0.42 (0.35–0.56) EU/mL, no MS: 0.34 (0.30–0.40) EU/mL, *p* = 0.009) (Figure 1, Table 2, Appendix A). LPS/HDL was also higher in MS vs. no MS patients (MS: 0.28 (0.24–0.38), no MS: 0.22 (0.17–0.26), *p* = 0.01) (Figure 2, Table 2, Appendix A). Most of the LPS/HDL and LPS deciles differed significantly between the groups in line with the median comparisons (Appendix A, Figure 1 and Figure 2). The levels of LPS-binding proteins and hsCRP did not differ between MS groups.

In comparison to controls, the T1D group had statistically significantly higher levels of hsCRP (control: 0.47 (0.26–0.75) mg/L, T1D: 0.84 (0.49–1.78) mg/L, *p* = 0.002) and EndoCAb IgG (control 71.1 (55.7–101.5) GMU/mL, T1D 92.6 (65.8–148.6) GMU/mL, *p* = 0.091) (Appendix A, Appendix A).

In the T1D group, only six patients had calprotectin levels above 50 µg/g, and five of them had calprotectin levels above 200 µg/g. Therefore, to reduce data variability, we considered calprotectin measurements above 200 µg/g as outliers for further statistical analysis.

Within the T1D group, patients with MS had statistically significantly higher calprotectin levels at middle deciles (quantile difference 40%: 4.3 (0.23, 9.98), *p* = 0.036; 50%: 6.9 (1.24, 11.81), *p* = 0.014; 60%: 8.72 (2.47,13.13), *p* < 0.011) (Figure 3), although the medians of calprotectin did not differ statistically significantly between the groups.

The median level of fecal calprotectin also did not differ between T1D and controls. However, patients with T1D had statistically significantly lower calprotectin levels in the first three deciles (quantile difference 10%: −2.14 (−3.34, −0.44), *p* = 0.017; 20%: −1.86 (−3.26, −0.40), *p* = 0.016; 30%: −1.54 (−3.21, 0.14), *p* = 0.071) (Appendix A, Appendix A).

### 3.3. Correlation Analysis in Study Groups

Patterns of correlations were compared between the T1D group and control group (data presented in Figure 4, Appendix A). In the T1D group, hsCRP correlated positively with EndoCAb IgG (r = 0.22, *p* = 0.07) and LBP (r = 0.36, *p* = 0.002). hsCRP did not correlate with any of the endotoxemia markers in the control group. Both in the control and diabetes groups, hsCRP correlated positively with waist/height ratio and NAFLD index HSI. hsCRP correlated positively with HbA1c (r = 0.5, *p* = 0.004) and BMI (r = 0.38, *p* = 0.034) only in the control group. The LPS and LPS/HDL ratio did not correlate with other inflammatory markers in the T1D group. In the control group, LPS correlated positively with LBP (r = 0.36, *p* = 0.045) and negatively with EndoCAb IgM (r = −0.41, *p* = 0.021), and the LPS/HDL ratio correlated positively with LBP (r = 0.39, *p* = 0.03). LPS correlated positively with FLI, as well as with serum triglycerides in both study groups. In the control group, it also correlated positively with leukocytes (r = 0.32, *p* = 0.075) and negatively with HDL (r = −0.032, *p* = 0.073). In the T1D group, LPS/HDL correlated positively with FLI (r = 0.23, *p* = 0.053) and triglycerides (r = 0.52, *p* < 0.001), and negatively with diabetes duration (r = −0.22, *p* = 0.069). In the control group, LPS/HDL ratio correlated positively with weight (r = 0.35, *p =* 0.05), ALT (r = 0.31, *p =* 0.087), GGT (r = 0.3, *p =* 0.097), leukocytes (r = 0.33, *p =* 0.062) and triglycerides (r = 0.67, *p =* 0 < 0.001), and negatively with eGDR (r = −0.32, *p =* 0.077). Similar to endotoxin, LBP correlated positively with leukocytes (r = 0.34, *p =* 0.057) in the control group, but not in the T1D group. Fecal calprotectin did not correlate with any of the inflammatory or endotoxemia markers in any of the groups.

### 3.4. Regression Analysis in T1D Group

The normality assumption was not rejected for log-transformed hsCRP in the T1D group (*p* = 0.134). The linear regression model results demonstrated that the significant predictors of log(hsCRP)) were LBP (β = 0.30 (0.09; 0.51), *p* = 0.005) and EndoCAb IgG (β = 0.29 (0.07; 0.50), *p* = 0.008); however, LPS/HDL (β = 0.19 (−0.03; 0.41), *p* = 0.084) and diabetes duration (β = 0.24 (−0.01; 0.49), *p* = 0.059) were significant at the 10% significance level. Overall, Model 1 was significant with F(7, 60) = 3.97 (*p =* 0.001), and 32% of the variability in data was explained by the fitted model (Table 3). 

Logistic regression model results demonstrated that the significant predictors of MS were EndoCAb IgM (OR = 0.32 (0.11; 0.93), *p* = 0.036), LPS/HDL (OR = 6.5 (2.1; 20.0), *p* = 0.001), diabetes duration (OR = 3.43 (1.21; 9.8), *p* = 0.021) and body mass index (OR = 2.3 (1.1; 4.9), *p* = 0.025). The Nagelkerke R2 of Model 2 was 0.56 and AIC = 72.8 (Table 3).

## 4. Discussion

In this work, we reported higher levels of endotoxemia in patients with T1D and MS, as compared to T1D subjects with no MS. In addition, we observed statistically significant associations between endotoxemia markers, hsCRP and MS in T1D. Finally, we described distinct relationships between LPS, its binding proteins, HDL and leukocytes in T1D, as compared to the control group, which might indicate an impaired response to endotoxins in T1D.

We demonstrated here that patients with T1D and MS had higher LPS and LPS/HDL levels, as compared to subjects with T1D and no MS. This is in line with the findings of the previous studies [9]. However, the levels of hsCRP and LPS-binding proteins did not differ between MS groups in our study. This can be explained by the course of events in the inflammatory reaction. Indeed, the rise in CRP starts at 4–6 h and peaks at 36 h after infection [42]. Therefore, CRP, LPS and LPS-binding protein levels measured in one sample might not correlate. Nevertheless, the results of the correlation and regression analysis demonstrated the link between increased CRP and endotoxemia in the T1D group in our study, which is in agreement with the published data [8,9,14,15,43].

We are the first to report a positive correlation of NAFLD index FLI with hsCRP, LPS and LPS/HDL ratio in T1D. These findings might indicate that similarly to the situation in NAFLD in other population groups [25], this condition is linked to endotoxemia in T1D. It seems that the observed correlation of LPS with triglycerides as a component of FLI is largely “responsible” for LPS correlation with FLI in our sample. Associations between triglycerides and fat consumption were found to be associated with LPS in the general population [44] and T1D previously [12].

To gain more insights into mechanisms of response to endotoxins in T1D, we compared correlation patterns of measured biomarkers in T1D and control groups. As a result, we identified a distinct response to endotoxemia in T1D compared to controls. For example, endotoxins in the control group correlated positively with leukocytes and negatively with HDL, possibly illustrating a “healthy” response to LPS. None of these correlations were observed in the T1D group. One of the functions of HDL is to scavenge and neutralize LPS [45]; therefore, individuals with higher HDL levels might have lower LPS concentrations. A rise in leukocyte number is a well-known body response to bacterial invasion [42]. Lack of the above correlations in the T1D group might be linked to differences in HDL size and functionality previously reported in T1D [21].

LBP, one of the acute-phase response proteins, is involved in LPS binding, and its levels are usually increased in endotoxemia along with changes in EndoCAb IgM and IgG levels [18,46]. Interestingly, we did not observe a correlation of LPS with any of these endotoxemia-response proteins in the T1D group. In contrast, in the control group, LPS and the LPS/HDL ratio correlated positively with LBP and negatively with EndoCAb IgM, as reported by others [19].

To summarize, the observed differences in response to endotoxins between T1D and control might be related to changes in the immune response in T1D [17].

In this study, we reported several novel approaches to statistical analysis. For example, in two-group comparisons, utilization of PP plots and quantile (or decile) difference plots helped to detect the group effect when the Wilcoxon test failed to do so. For example, in the T1D group stratified according to MS presence, median calprotectin levels did not differ significantly between groups according to the Wilcoxon test (*p =* 0.18). However, there were significant differences at the three middle deciles (40th, 50th and 60th percentiles) according to the EL test. The explanation of the insufficient power of the Wilcoxon test to detect these effects can be based on the skewness of the data distributions and observations relatively far from the median. For example, in T1D patients, we can see from the one-dimensional scatterplot (Figure 3b) that seven observations of calprotectin were relatively far from the central location of distribution in patients without MS. Also, from the kernel density plots, we can see that robust statistical methods should be used for some further investigations. As another example, according to the Wilcoxon test, calprotectin medians did not differ significantly between the control and T1D. However, the variance for the T1D group was much higher (it was confirmed by the Lepage test), leading to big differences at lower and higher quantiles. The variance difference can be seen from the quantile difference plot and PP plot. Thus, more detailed statistical analysis using different data visualization tools and additional tests to compare lower and higher deciles or quantiles should give us a better understanding of the differences between two samples [33].

Our study’s limitations include a relatively low number of subjects and the cross-sectional nature of this study, which did not allow us to evaluate the causal relationship between MS, markers of endotoxemia and hsCRP in T1D.

The major strength of the study is the analysis of the LPS, LBP, EndoCAb IgM, EndoCAb IgG and fecal calprotectin in patients with T1D, in comparison to healthy subjects and when stratified according to MS status, which has not been reported previously and allowed us to identify signs of a distinct response to endotoxins in T1D. Finally, the non-traditional approaches of statistical analysis presented in this paper might be useful for other researchers in biomedicine for gaining a deeper understanding during two-sample comparisons.

## 5. Conclusions

To conclude, we report higher levels of endotoxemia in subjects with T1D and MS and statistically significant associations between endotoxemia markers, hsCRP and MS in T1D. We also observed distinct patterns of response to LPS in T1D, as compared to the control group. These findings are important for clinical practice as they underline the added value of screening and treatment of MS as a state predisposing to endotoxemia. On the other hand, our data warrant more research on inflammatory response mechanisms in T1D, which might lead to the development of novel treatment options for low-grade inflammation and MS to delay the progression of T1D complications.

## Figures and Tables

**Figure 1 biomedicines-11-03269-f001:**
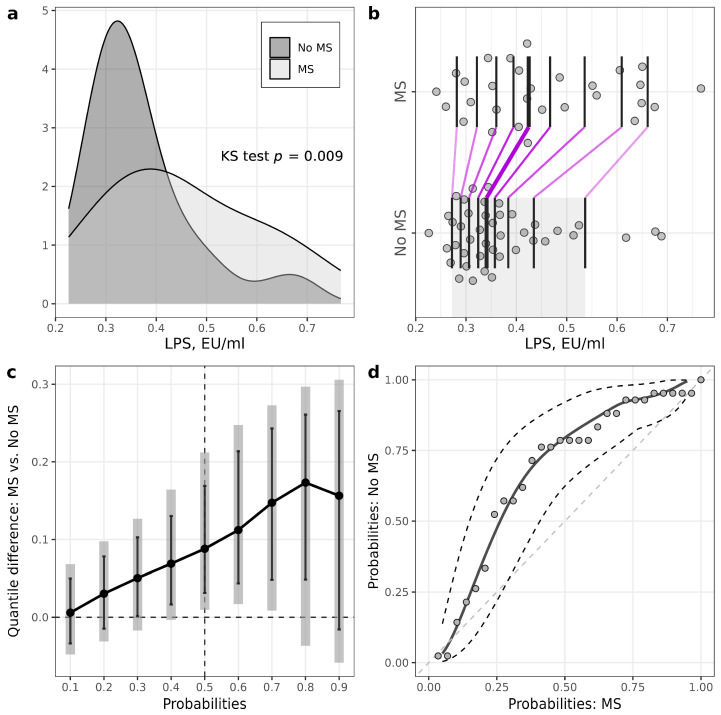
Illustrations for comparison of LPS levels in patients with T1D stratified according to the presence of metabolic syndrome (MS). (**a**) Plots of densities in patients with and MS, with *p*-value of Kolmogorov–Smirnov (KS) test for the equality of distributions; (**b**) one-dimensional scatterplots for patients with and without MS, with vertical bars representing deciles in each group (R library rogme); (**c**) quantile difference (MS—no MS) plot constructed at deciles, with estimates obtained using two-sample smoothed empirical likelihood method (R library EL), error bars representing 95% pointwise CIs and shaded bars representing 95% simultaneous confidence bands for 9 deciles; (**d**) probabilities of patients with MS versus probabilities of patients without MS (PP-plot) with estimates and 95% pointwise CIs obtained using the two-sample smoothed empirical likelihood method (R library EL). Bandwidths for kernel estimation in (**c**,**d**) were selected using the method of Sheather and Jones (function bw.SJ in R).

**Figure 2 biomedicines-11-03269-f002:**
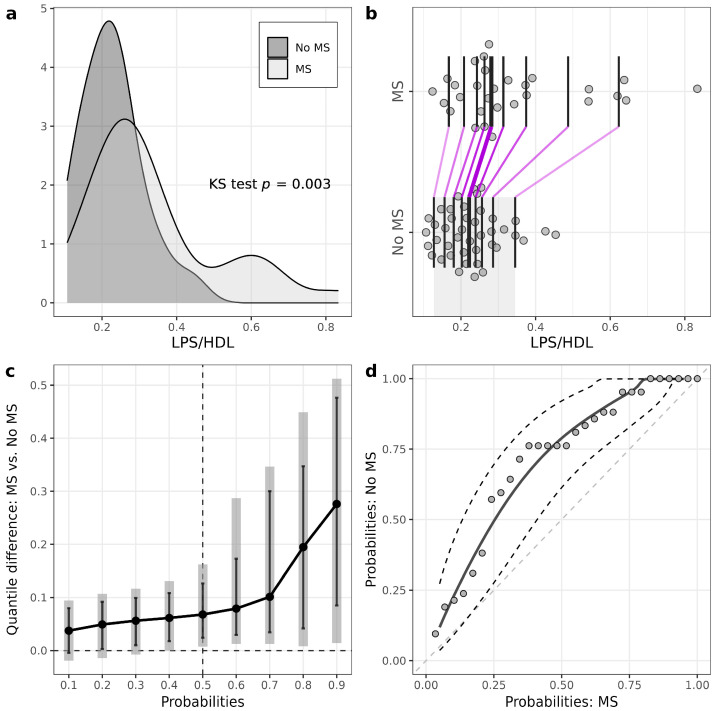
Illustrations for comparison of LPS/HDL in patients with T1D stratified according to the presence of metabolic syndrome (MS). (**a**) Plots of densities in patients with and without MS, with *p*-value of Kolmogorov–Smirnov (KS) test for the equality of distributions; (**b**) one-dimensional scatterplots for patients with and without MS, with vertical bars representing deciles in each group (R library rogme); (**c**) quantile difference (MS — No MS) plot constructed at deciles, with estimates obtained using two-sample smoothed empirical likelihood method (R library EL), error bars representing 95% pointwise CIs and shaded bars representing 95% simultaneous confidence bands for 9 deciles; (**d**) probabilities of patients with MS versus probabilities of patients without MS (PP-plot) with estimates and 95% pointwise CIs obtained using the two-sample smoothed empirical likelihood method (R library EL). Bandwidths for kernel estimation in (**c**,**d**) were selected using the method of Sheather and Jones (function bw.SJ in R).

**Figure 3 biomedicines-11-03269-f003:**
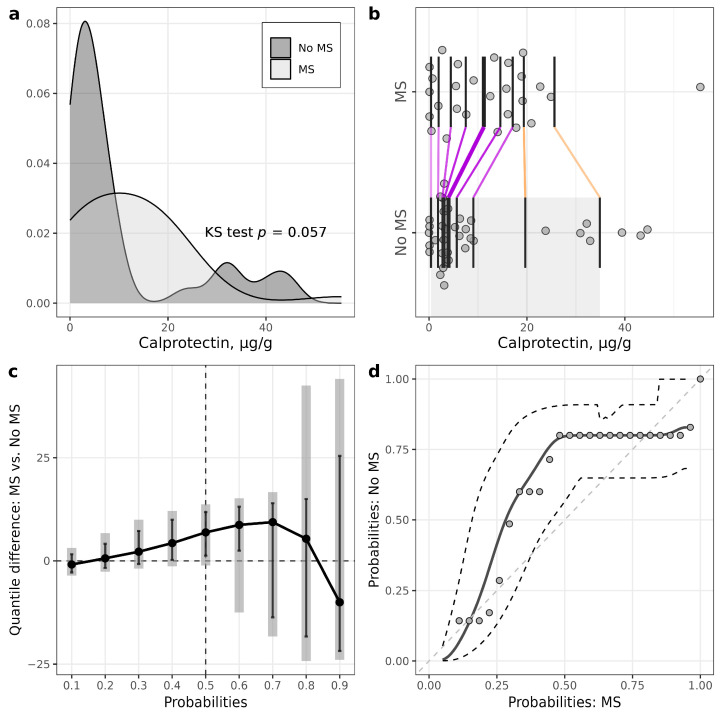
Illustrations for comparison of calprotectin in patients with T1D stratified according to the presence of metabolic syndrome (MS). (**a**) Plots of densities in patients with and without MS, with *p*-value of Kolmogorov–Smirnov (KS) test for the equality of distributions; (**b**) one-dimensional scatterplots for patients with and without MS, with vertical bars representing deciles in each group (R library rogme); (**c**) quantile difference (MS—No MS) plot constructed at deciles, with estimates obtained using two-sample smoothed empirical likelihood method (R library EL), error bars representing 95% pointwise CIs and shaded bars representing 95% simultaneous confidence bands for 9 deciles; (**d**) probabilities of patients with MS versus probabilities of patients without MS (PP-plot), with estimates and 95% pointwise CIs obtained using two-sample smoothed empirical likelihood method (R library EL). Bandwidths for kernel estimation in (**c**,**d**) were selected using the method of Sheather and Jones (function bw.SJ in R).

**Figure 4 biomedicines-11-03269-f004:**
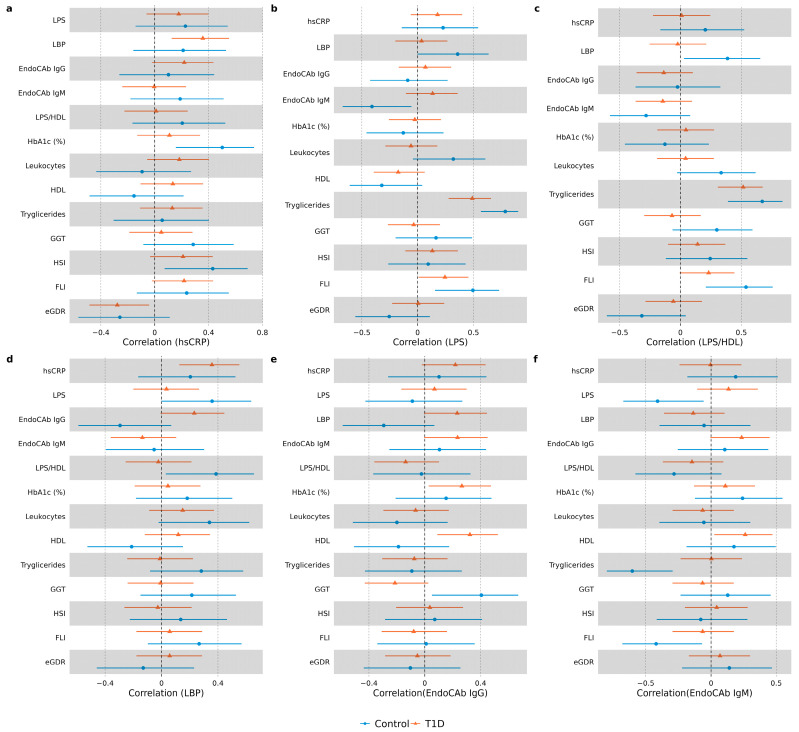
Forest plots demonstrating correlations between hsCRP and endotoxemia markers in each study group. Spearman correlation coefficients (95% CI) are illustrated with forest plots for hsCRP (**a**), LPS (**b**), LPS/HDL (**c**), LBP (**d**), EndoCAb IgG (**e**) and EndoCAb IgM (**f**).

**Table 1 biomedicines-11-03269-t001:** Characteristics of subjects.

Phenotype	No Metabolic Syndrome (*n* = 43)	Metabolic Syndrome (*n* = 31)	*p*
Male/Female, *n* (%)	19/24 (44/56)	9/22 (29/71)	0.28
Age, years	38 (32–50)	45 (36–54)	0.065
BMI, kg/m^2^	23.5 (21.45–25.65)	28 (25.4–32.65)	<0.001
Waist/height ratio	0.457 (0.431–0.497)	0.56 (0.509–0.621)	<0.001
Smoker, *n* (%)	11 (26)	9 (29)	0.95
Hypertension, *n* (%)	15 (35)	20 (65)	0.022
Length of diabetes, years	20 (11–32)	24 (17–34)	0.062
Retinopathy, *n* (%)	13 (30)	19 (61)	0.015
CVD, *n* (%)	3 (7)	6 (19)	0.21
Macroalbuminuria, *n* (%)	2 (5)	5 (16)	0.21
On ACEI/ARB, *n* (%)	4 (9)	11 (35)	0.013
On lipid-lowering medication, *n* (%)	5 (12)	12 (39)	0.014
Autoimmune thyroid disease, *n* (%)	9 (21)	10 (32)	0.41
Hemoglobin A1C, %	7.3 (6.8–9.0)	8.4 (7.4–9.3)	0.087
Hemoglobin A1C, mmol/mol	53 (47–67)	64 (54–64)	0.087
Estimated glomerular filtration rate, mL/min/1.73m^2^	114 (104–119)	98 (74–108)	<0.001
Albumin/creatinine ratio in urine, mg/mmol	0.38 (0.19–1.05)	0.97 (0.37–2.34)	0.072
Total cholesterol, mmol/L	4.72 (4.29–5.18)	5.51 (4.89–6.04)	0.006
Low-density lipoproteins, mmol/L	2.68 (2.08–3.24)	3.22 (2.47–3.75)	0.02
High-density lipoproteins, mmol/L	1.71 (1.38–2.01)	1.44 (1.21–1.73)	0.012
Triglycerides, mmol/L	1.02 (0.79–1.18)	1.72 (1.30–2.11)	<0.001
Alanine aminotransaminase, U/L	19 (134–26)	21 (17–28)	0.087
Aspartate aminotransferase, U/L	22 (19–28)	25 (17–31)	0.879
Gamma-glutamyltransferase, U/L	15 (13–22)	18 (15–26)	0.212
Bilirubin, µmol/L	10.3 (8.0–13.1)	8.4 (6.3–10.6)	0.015
Hemoglobin, g/L	141 (134–149)	139 (126–150)	0.321
Erythrocytes, 10 × 12/L	4.7 (4.5–4.9)	4.7 (4.4–5.1)	0.583
Leukocytes, 10 × 9/L	6.2 (5.2–7.5)	6.1 (5.0–7.1)	0.576
Thrombocytes, 10 × 9/L	262 (238–280)	254 (226–295)	0.518
eGDR	4.9 (2.8–6.7)	2.1 (1.0–3.3)	0.001
FLI	13.2 (6.0–22.1)	54.0 (30.1–78.9)	<0.001
HSI	31.2 (28.8–33.7)	39.6 (34.5–42.0)	<0.001

For categorical variables, data are presented as counts (%), and the equality of proportions between study groups was tested using the chi-square test for the equality of proportions. For continuous variables, data are presented as medians (IQR), and the comparisons between the control and T1D group were performed using the Wilcoxon test. BMI—body mass index. CVD—cardiovascular disease. ACEI—angiotensin-converting enzyme inhibitors. ARB—angiotensin receptor blockers. eGDR—estimated glucose disposal rate. FLI—fatty liver index. HSI—hepatic steatosis index.

**Table 2 biomedicines-11-03269-t002:** Inflammatory marker analysis in patients with T1D stratified by metabolic syndrome.

Phenotype	No Metabolic Syndrome (*n* = 43)	Metabolic Syndrome (*n* = 31)	*p*
hsCRP, mg/L	0.80 (0.49–1.53)	1.23 (0.38–2.17)	0.41
LPS, EU/mL	0.34 (0.30–0.40)	0.42 (0.35–0.56)	0.009
LPS/HDL ratio	0.22 (0.17–0.26)	0.28 (0.24–0.38)	0.001
EndoCAb IgG, GMU/mL	89.5 (67.1–150.2)	96.0 (59.1–141.6)	0.84
EndoCAb IgM, MMU/mL	46.6 (34.2–85.2)	43.7 (31.3–60.4)	0.33
LBP, µg/mL	11.1 (7.9–16.3)	11.1 (7.9–13.3)	0.89
Calprotectin, µg/g	3.8 (2.7–8.6)	12.5 (3.2–18.4)	0.18

Data are presented as medians (IQR). *p*-values of the Wilcoxon test between the study groups are presented. The results did not differ significantly after adjusting for sex, diabetes duration and BMI (ANCOVA on ranks).

**Table 3 biomedicines-11-03269-t003:** Regression analysis in type 1 diabetic patients.

Model 1			Model 2		
Predictor	β	*p*	Predictor	OR (95% CI)	*p*
LBP	0.30 (0.09; 0.51)	0.005	LBP	0.92 (0.45; 1.90)	0.82
EndoCAb IgG	0.29 (0.07; 0.50)	0.008	EndoCAb IgG	1.67 (0.81; 3.45)	0.16
EndoCAb IgM	−0.06 (−0.27; 0.16)	0.61	EndoCAb IgM	0.32 (0.11; 0.93)	0.036
LPS/HDL	0.19 (−0.03; 0.41)	0.084	LPS/HDL	6.5 (2.1; 20.0)	0.001
hsCRP	-	-	hsCRP	0.57 (0.26; 1.25)	0.16
Sex (female)	0.23 (−0.22; 0.68)	0.31	Sex (female)	2.7 (0.6; 12.7)	0.21
Diabetes duration	0.24 (−0.01; 0.49)	0.059	Diabetes duration	3.4 (1.2; 9.8)	0.021
BMI	0.10 (−0.13; 0.32)	0.39	BMI	2.3 (1.1; 4.9)	0.025
R^2^ = 0.32, F(7, 60) = 3.97, *p =* 0.001			AIC = 72.8, Nagelkerke R^2^ = 0.56		

Model 1—multivariate linear regression for log-transformed hsCRP as the response variable in T1D group, where data are presented as coefficient estimates with 95% CI; Model 2—multivariate logistic regression for the presence of metabolic syndrome as the response variable in T1D group, where data are presented as odds ratio (OR) with 95% CI. All predictors were standardized before model fitting, and multicollinearity was examined using variance inflation factor calculations.

## Data Availability

The data are not available publicly due to restrictions of privacy, but they are available from the corresponding author on request.

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
