# Peer review of "Association of Endotoxemia with Low-Grade Inflammation, Metabolic Syndrome and Distinct Response to Lipopolysaccharide in Type 1 Diabetes"

_biomedicines, 2023, doi:10.3390/biomedicines11123269_

Round 1

Reviewer 1 Report

Comments and Suggestions for Authors

Authors try to find out correlation between  endotoxaemia with metabolic syndrome (MS) and low grade inflammation 10 in type 1 diabetes (T1D). The major limitation of presented studies is small amount of samples and only in one point time collections. The level of endotoxins is similar in control 33 subject and patients 74. The level of LPS, EU/ml is equal to 0.31 for both cases (see Tab. 2) It is no evidence that patients suffer from edotoxemia. It look like that Authors try to correlate all laboratory data. For example Table 1 line 1 male gender P 0.81 – is it any voluble results ?

Data for statistical analysis were examined by “ Data on five serum inflammatory markers was  examined for the homogeneity across three plates. Initial visual inspection indicated 158 highly skewed distributions”  Did visual inspection might be good method?

Authors do not consider heterogeneity of tested samples – male/female, longitude of T1D, medical treatment, history of MS occurrence.

Valuable part of manuscript is extensive statistical analysis of routine diagnostic data. However, if statistical validation methods is major goal of studies another journal is recommended.

In current state of manuscript presented results are to preliminary to be published.

Author Response

Response to Reviewer 1.

We thank you for taking the time to review our manuscript. We have addressed all issues raised, which helped to make the manuscript more focused to the associations between endotoxaemia and metabolic syndrome in type 1 diabetes. We hope very much that the corrections made the manuscript more comprehensive and suitable for publication.

  1. The major limitation of presented studies is small amount of samples and only in one point time collections.

We sincerely appreciate the reviewer's thoughtful consideration of our study. We acknowledge the limitations associated with our research, including the relatively small sample size and the use of one-time data collection, as pointed out. These limitations are duly noted in the discussion section of the manuscript and we recognize their impact on the generalizability of our findings. However, we would like to underline that we present analysis of high range of endotoxaemia markers (not only LPS but also its binding proteins), which are not performed in clinical everyday practice. Moreover, as we outline in the introduction, data on endotoxaemia in T1D are scarce in the literature, therefore our data are novel and add to the current knowledge.

  1. The level of endotoxins is similar in control 33 subject and patients 74. The level of LPS, EU/ml is equal to 0.31 for both cases (see Tab. 2) It is no evidence that patients suffer from edotoxemia.

Regarding the observed similarity in the level of endotoxins between the control and T1D patient groups, we agree that this observation, taken in isolation, might suggest that patients do not suffer from endotoxemia. However, it is essential to consider that our study involved a more detailed analysis of the data. After stratifying the T1D patients based on the presence of Metabolic Syndrome (MS), we observed that patients with T1D and MS exhibited a higher level of LPS and LPS/HDL ratio compared to those without MS. This specific finding, as detailed in Figure 1 and 2, Table 2 and Supplement Table 2 and 3 of the revised manuscript version, highlights a significant difference in endotoxaemia levels based on the presence of MS. These results provide important insights into the relationship between endotoxemia, metabolic syndrome, and T1D.

Thanks to your comment we understood that the initial manuscript version lacked sufficient focus on the problem of endotoxaemia associated with MS in T1D. In the revised version, we put the focus on comparison between MS groups within T1D and mention comparisons between T1D and control only briefly. To achieve this, characteristics of subjects (Table 1) are now presented for T1D patients stratified according to MS. Table with characteristics of T1D versus control group is moved to supplements (supplemental table 1 in the revised manuscript version). We now also present data on endotoxaemia markers in MS and non-MS groups in the table in the main manuscript text (Table 2). Data on endotoxaemia marker comparisons between T1D and controls is also moved to supplements (supplemental Table 4), along with corresponding figures.

  1. It looks like Authors try to correlate all laboratory data. For example Table 1 line 1 male gender P 0.81 – is it any voluble results?

The analysis in Supplemental Table 1 (Table 1 in the initial manuscript version) aims to provide a comprehensive and descriptive understanding of the demographic and clinical characteristics of the study subjects in both the control and T1D groups. Categorical variables are presented as counts (%) and the equality of proportions between the control and T1D groups was assessed using the chi-square test for equality of proportions (as stated below the table). The results in line 1 indicate that the proportions of males and females were approximately equal in both study groups (p=0.81). Meanwhile, we also demonstrate the categorical variables which have significant association with the study groups (for example, line 5 indicates that metabolic syndrome (MS) was more frequent in the T1D group compared to the Control group, p=0.005).

However, for better readability we changed line 1 in Table 1 in the following way:

Male gender, N (%) 

14 (42.4 %)

28 (37.8 %)

0.81

Male/Female, N (%)

14/19 (42.4/57.6 %)

28/46 (37.8/62.2 %)

0.81

In the main manuscript body, we placed the Table 1 with information on characteristics of patients within T1D group stratified according to MS (also see explanation above).

  1. Data for statistical analysis were examined by “Data on five serum inflammatory markers was examined for the homogeneity across three plates. Initial visual inspection indicated highly skewed distributions”. Did visual inspection might be good method?

While visual inspection alone may not be considered a definitive method, we believe it is a crucial initial step in data analysis. We noted that the visual inspection was used as a preliminary check, followed by the normality test of the five inflammatory markers (described in Statistical analysis section of the manuscript). It was also important to determine if the distributions are not symmetric or at least close to the normal distribution to choose the most appropriate methods for the statistical analysis. However, taking into account the reviewer’s inquiry, we changed the text accordingly without mentioning explicitly the visual inspection in the description of the statistical methods used.

"Data on five serum inflammatory markers was examined for the homogeneity across three plates. The distributions were highly skewed to the left (skewness coefficient from 0.66 to 1.93); normality assumption was violated for all five markers (p<0.001)."

  1. Authors do not consider heterogeneity of tested samples – male/female, longitude of T1D, medical treatment, history of MS occurrence.

In the two-sample comparisons (between groups of T1D and Control, as well as between T1D patient groups stratified by MS), we utilized the Wilcoxon test for locations and the empirical likelihood method for quantiles. The analysis presented in Table 1 of the initial manuscript version (currently supplemental Table 1) is purely descriptive, and thus, adjustment variables were not considered.

In response to the reviewer's comment, we extended our analysis in of the revised manuscript version by performing ANCOVA on ranks to adjust for sex, diabetes duration, and BMI additionally to Wilcoxon test (Table 2 and Supplemental Table 4). However, one limitation of employing ANCOVA on ranks is the somewhat intricate interpretation of the results, as comparisons are based on estimated marginal mean ranks. 

Regarding the quantile comparisons using the empirical likelihood method, to our knowledge, there are no available procedures for covariate adjustment yet.

In the regression analysis within the T1D group (see Table 3), our models were adjusted for sex and diabetes duration. For consistency, we also included BMI in both models as the covariate in the revised manuscript.

  1. Valuable part of manuscript is extensive statistical analysis of routine diagnostic data. However, if statistical validation methods is major goal of studies another journal is recommended.

Our primary aim in the statistical analysis of this study was not only to provide comprehensive and refined data analysis but also to utilize advanced statistical methods to uncover subtle effects that may be present in the data. For instance, while the classical Wilcoxon test, as shown in Supplemental Table 4, line 7, does not reveal a difference in median calprotectin levels between the control and T1D groups, the use of additional tests, along with visualizations, suggests that calprotectin variances between both study groups are significantly different (see also supplemental Figure 4). We are grateful to the reviewer for their comments and suggestions. We hope that our revisions effectively address the reviewer's concerns, and the study results, along with the incorporation of novel approaches in routine diagnostic data analysis, are suitable for publication in this journal.

  1. In current state of manuscript presented results are too preliminary to be published.

During the revision process we have performed extensive work to make the manuscript more comprehensive and to underline its added value. Mainly, we have changed the text and placement of tables and figures, putting all data referring to association between MS and endotoxaemia in T1D in the main manuscript body. Two group comparison between T1D and controls are placed to supplements. We did not exclude the data on control group from the manuscript completely as we wanted to leave correlation analysis presented both in T1D and control group, to underline distinct response patterns to endotoxaemia between T1D and generally healthy subjects. The discussion section was also changed – we omitted data which is interesting but not that much referring to the manuscript topic.

We think that data presented in our manuscript are valuable and deserve publication. As mentioned before, we did not find published data on several endotoxaemia markers (not only LPS but also its binding proteins) in association with metabolic syndrome in T1D in the literature. Our findings are important for clinical practice as they underline the added value of screening and treatment of MS as a state predisposing to endotoxaemia in T1D. We of course understand that our results, although deserving publication, can be called preliminary and warrant more research on inflammatory response mechanisms in T1D. In the future, this new knowledge might lead to the development of novel treatment options for low-grade inflammation and MS to delay progression of T1D complications.

Reviewer 2 Report

Comments and Suggestions for Authors

This study reported on the correlation with levels of LPS the LPS binding protein (LBP), EndoCAb, IgG and IgM and hsCRP in patients with various forms of MS and compared to control patients without MS. Their findings support previous research, as they mention, although hsCRP and LPS binding proteins levels did not differ between the MS groups. The authors state that their samples sizes maybe limiting  but they have a conducted a thorough investigation with the samples they had and performed various analyses to examine if correlations existed. The authors presented all parameters used and details enough that others will be able to build off of their findings and potentially even try other types of analyses on the same data sets.

In my view, it is important  to be able to repeat studies and replicate previous reports as well as present data which may show differences from earlier studies as then follow up studies can be conducted. Such subtle effects of the populations used in one study may be different in ethnic or genetic background to account for differences,  or even dietary differences. Himan studies as such are difficult to conduct due to variability in factors which are hard to control for.

In my view, the findings of this study are beneficial for the health care community as well as researchers in health care. The studies are detailed in presentation and analysis. The manuscript is well written, and the authors are cautious on not overstating any findings.

In reading, I did not find any edits required. The text is dense in analysis and hard for me to follow on each statistical approach. A biostatistician reviewer may note a suggestion on a topic which I might have missed. The authors apparently included a well qualified biostatistician for this study and presented the data well.

Author Response

Response to Reviewer 2.

Thank you for taking the time to review our manuscript. We are grateful for your encouraging remarks and are pleased to know that our research findings have met the standards for publication.

Comment: This study reported on the correlation with levels of LPS the LPS binding protein (LBP), EndoCAb, IgG and IgM and hsCRP in patients with various forms of MS and compared to control patients without MS. Their findings support previous research, as they mention, although hsCRP and LPS binding proteins levels did not differ between the MS groups. The authors state that their samples sizes maybe limiting  but they have a conducted a thorough investigation with the samples they had and performed various analyses to examine if correlations existed. The authors presented all parameters used and details enough that others will be able to build off of their findings and potentially even try other types of analyses on the same data sets.

In my view, it is important  to be able to repeat studies and replicate previous reports as well as present data which may show differences from earlier studies as then follow up studies can be conducted. Such subtle effects of the populations used in one study may be different in ethnic or genetic background to account for differences,  or even dietary differences. Himan studies as such are difficult to conduct due to variability in factors which are hard to control for.

In my view, the findings of this study are beneficial for the health care community as well as researchers in health care. The studies are detailed in presentation and analysis. The manuscript is well written, and the authors are cautious on not overstating any findings.

In reading, I did not find any edits required. The text is dense in analysis and hard for me to follow on each statistical approach. A biostatistician reviewer may note a suggestion on a topic which I might have missed. The authors apparently included a well qualified biostatistician for this study and presented the data well.

Reply: We want to note that due to the comments of reviewer 1, we had to address the issues he raised. This helped to make the manuscript more focused on the associations between endotoxaemia and metabolic syndrome in type 1 diabetes. We hope very much that due to the corrections the manuscript became more comprehensive and easier to read.

Round 2

Reviewer 1 Report

Comments and Suggestions for Authors

On revised manuscript version Authors do not over-come major weaknesses of presented studies – low amount of samples and their heterogeneity. I suggested that Authors pointed out that weaknesses on Abstract and Discussion part of manuscript, avoided misleading readers.

Author Response

Response to Reviewer 1.

We thank you for taking the time to review our manuscript. We have addressed your comments. We hope very much that the manuscript is suitable for publication now

  1. On revised manuscript version Authors do not over-come major weaknesses of presented studies – low amount of samples and their heterogeneity. I suggested that Authors pointed out that weaknesses on Abstract and Discussion part of manuscript, avoided misleading readers.

We show clearly the number of subjects in the abstract, in results section, in all tables. We also discuss this issue in the discussion section. As for heterogeneity, we performed ancova analysis which did not demonstrate changes in the results after adjustment for major confounders. Therefore, we are sure that we have done everything to present our results in unbiased way. We have also extensively explained in reply to the first round comments.
